# Enhancing Quality Control in Web-based Participatory Augmented Reality Business Card Information System Design

**DOI:** 10.3390/s23084068

**Published:** 2023-04-18

**Authors:** Yongjun Kim, Yung-Cheol Byun

**Affiliations:** 1Department of Computer Engineering, Jeju National University, Jeju 63243, Republic of Korea; 2Department of Computer Engineering, Major of Electronic Engineering, Institute of Information Science & Technology, Jeju National University, Jeju 63243, Republic of Korea

**Keywords:** augmented objects, augmented reality, business card information, internet, location information, quality control

## Abstract

The rapid development of information and communication technology has fostered a natural integration of technology and design. As a result, there is increasing interest in Augmented Reality (AR) business card systems that leverage digital media. This research aims to advance the design of an AR-based participatory business card information system in line with contemporary trends. Key aspects of this study include applying technology to acquire contextual information from paper business cards, transmitting it to a server, and delivering it to mobile devices; facilitating interactivity between users and content through a screen interface; providing multimedia business content (video, image, text, 3D elements) via image markers recognized by users on mobile devices, while also adapting the type and method of content delivery. The AR business card system designed in this research enhances traditional paper business cards by incorporating visual information and interactive elements and automatically generating buttons linked to phone numbers, location information, and homepages. This innovative approach enables users to interact and enriches their overall experience while adhering to strict quality control measures.

## 1. Introduction

A business card is a professional introduction, displaying one’s name, address, phone number, company name, and title. As a crucial marketing tool for establishing connections between businesses and potential customers, business cards represent a company’s brand and its products. Traditional paper business cards aim to make a strong first impression during business meetings, emphasizing simplicity and conveying essential information. As a result, unique and differentiated design business cards and digital business cards have emerged [1].

The rapid development of information and communication technology has led to the rise of digital science, technology museums, and exhibitions featuring virtual books, digital Olympic halls, voice recognition robots, virtual architectural expressions, interactive illusions, and camera games [2]. Smartphones, offering convenience and PC-like functionality, have gradually replaced conventional mobile phones. However, only a few applications currently utilize Augmented Reality (AR), despite the growing demand for target recognition and navigation functions in environments with limited wireless networks [3].

Today, characterized by the fourth industrial revolution, the proliferation of mobile devices has resulted in better connectivity and segmentation between customers compared to traditional mass marketing and internet marketing. Mobile marketing has emerged as a popular tool for targeted marketing, with companies developing branded applications and leveraging mobile device capabilities for various marketing activities [4]. Consequently, mobile business card services operating on mobile platforms have started to appear, facilitating information exchange, multimedia content sharing, and diverse communication options.

Paper business cards, though widely used, face limitations in conveying information in the digital age and have a restricted role in public relations. AR business cards, which employ AR technology, can address these challenges. AR technology involves object recognition and location-tracking capabilities, as exemplified by the popular AR game “Pokémon Go” which utilizes location-tracking technology [5]. The widespread use of smartphones, which are equipped with GPS, cameras, and displays, has contributed to the growing interest in AR, as these devices are ideal for realizing their potential. Additionally, smartphones enable real-time information exchange while moving, facilitating high-level communication [6].

Augmented Reality technology merges the natural environment with virtual space in real time, providing users with an immersive and realistic experience. Users can access information intuitively by overlaying information on actual locations, enhancing their engagement and interaction [7]. Recent advancements have seen the integration of AR technology into smartphones, with the development of business card applications that display animations and pre-rendered 3D graphics [8]. This research aims to capitalize on rapidly changing trends and cutting-edge information and communication technology to revolutionize marketing and business by reimagining business cards.

Quality control is essential to many industries and products, ensuring businesses deliver consistent and reliable goods and services to their customers [9]. Business cards are no exception, as they serve as the first point of contact in marketing and networking efforts, representing individuals and businesses in a physical or digital format. The quality of business cards can significantly impact the image and perception of a company or an individual. Therefore, it is crucial to explore how advancements in technology, such as Augmented Reality (AR), can enhance the quality control of business cards in terms of design, production, and user experience.

## 2. Related Research

### 2.1. Augmented Reality

Augmented Reality (AR) is technology that utilizes video devices to recognize objects in the real world from a human perspective. It seamlessly integrates the real and virtual worlds by displaying virtual information, such as text, as well as 2D and 3D objects, in real time on the recognized objects [10]. AR can be classified as location-based AR, marker-based AR, and non-marker-based AR, depending on the method used to identify real-world objects.

#### 2.1.1. Location-Based Augmented Reality

The widespread use of mobile devices, such as smartphones, has led to the emergence of location-based AR content. Location-based AR leverages mobile device GPS (Global Positioning System) data to measure the user’s latitude, longitude, and altitude. It uses geomagnetic sensors to determine the user’s direction and accelerometers to establish the device’s orientation. However, GPS sensors struggle to accurately calculate the user’s position indoors or underground, limiting their functionality to navigation pointing towards the target location [11].

#### 2.1.2. Marker-Based Augmented Reality

Marker-based AR utilizes square markers with high recognition rates to calculate the position and orientation of the camera, as video processing technology is significantly affected by external factors such as lighting and shading. Markers enable the efficient creation of coordinate systems for the real and virtual worlds by selecting classes to extend 3D objects. Marker-based AR processes the original image into a binarized image using a threshold, recognizes the marker’s edge to determine its orientation and position, and then augments objects accordingly [11].

#### 2.1.3. Non-Marker-Based Augmented Reality

Traditional AR technology relies on markers to augment virtual objects, requiring users to possess these features. The presence of markers in the output video may conflict with the real world, reducing user immersion and realism. To address these challenges, research is being conducted to determine the camera’s orientation based on natural environmental information in the real world. This approach, known as non-marker-based AR technology, has already led to commercial services [12,13].

AR technology offers the advantage of synthesizing and displaying virtual information in real space rather than merely providing virtual content within a virtual background. This expands users’ interaction capabilities and enhances their implicit knowledge. As a result, the technology holds significant potential for increasing utility value, enabling skill acquisition, and creating new high-value opportunities through integration with other industries [14].

Various industrial sectors are exploring new content services incorporating AR technology across different content, platform, network, and device (CPND) ecosystem levels [14]. Figure 1 depicts Ford’s AR repair support software, and Figure 2 showcases Movicon’s CE Overview [11].

### 2.2. Digital Business Card

Business cards primarily promote business success, products, brands, and users. Offline business cards fulfill this purpose and are generally easier to use than digital alternatives, such as mobile business cards. The rapid development of information and communication technology has led to a natural combination of technology and design, resulting in an increased appreciation for analog sensibility. This trend has driven the demand for offline business cards.

Moreover, the emergence of various content-providing services has created a growing need for alternative information delivery methods, such as digital product catalogs, brand advertisement introductions, and location information provision through offline business cards.

Recently, tangible media technology has gained traction, providing users with virtual experiences that resemble reality by bringing virtual information directly into the real world [17]. This technology aims to maximize user satisfaction by conveying all sensory information from a given scene, ensuring immersion and realism. Among experiential media technologies, Augmented Reality (AR) technology combined with historical content storage and distribution has gained significant domestic and international attention [18].

AR technology enables the display of virtual information data consistent with real-world experiences, blending reality and the virtual world. It involves the real-time synthesis of information with virtual objects based on reality [19,20]. AR technology has been employed in various fields by combining virtual environments with the natural world and providing visual interaction, stimulating curiosity and engagement [21,22]. In response to changing times, this research aims to integrate AR technology with sensory business cards to innovate business cards, a crucial element in PR and marketing for business success.

Figure 3 illustrates the differences and progression between paper and digital business cards. Some users find registration and management functions of mobile business card software challenging and continue to rely on offline business cards (paper business cards). Print volumes continue to grow, reflecting this trend. Offline business cards, being true to their purpose, hold an advantage in ease of use compared to digital business cards. However, the integration of technology and design has sparked interest in AR business card systems based on digital media.

This increased interest is due to the diversification of mobile AR platforms resulting from the widespread use of smartphones. The advancement of core technologies, such as the Internet of Things (IoT), computer vision, realistic content, and artificial intelligence, has led to the development of AR and the selective provision of reliable information. Consequently, expectations for utilizing AR technology in various business fields have risen.

### 2.3. Quality Control in AR Business Cards

Augmented Reality (AR) is technology that utilizes video devices to recognize objects in the real world from a human perspective. It seamlessly integrates the real and virtual worlds by displaying virtual information, such as text, as well as 2D and 3D objects, in real time on the recognized objects [10]. AR can be classified into location-based AR, marker-based AR, and non-marker-based AR, depending on the method used to identify real-world objects.

#### 2.3.1. Design Quality

The design quality of AR business cards plays a significant role in effectively communicating information and capturing users’ attention. Focusing on clarity, creativity, and relevance is essential to ensure high-quality design. Clarity refers to the legibility of text, discernibility of graphics, and unambiguous presentation of information. Creativity is crucial for capturing attention and differentiating the card from competitors, involving unique and innovative uses of AR features that highlight the brand identity. Relevance entails ensuring that the AR content aligns with the target audience’s interests and expectations while fulfilling the primary purpose of a business card.

#### 2.3.2. Production Quality

Quality control when producing AR business cards involves robustness, compatibility, and seamless integration. Robustness pertains to the reliability and resilience of the AR technology to perform under varying conditions, such as lighting, camera angles, and environments. Compatibility addresses the ability of the AR business card system to function across different devices and platforms, maximizing accessibility for users. Seamless integration focuses on creating a smooth and intuitive user experience that effectively merges the physical or digital business card with AR content.

#### 2.3.3. User Experience Quality

Ensuring a high-quality user experience is vital to the success of AR business cards. This involves ease of use, engagement, and effectiveness. Ease of use pertains to the simplicity and intuitiveness of the AR business card system, allowing users to access and interact with AR content without unnecessary complexity. Engagement refers to the level of interest and interaction the AR content elicits, driving users to explore further or to act (e.g., contacting the business). Effectiveness measures the extent to which the AR content achieves its intended outcome, such as promoting a brand, conveying information, or establishing connections. By focusing on these aspects of user experience quality, AR business cards can provide value and contribute to successful business relationships.

## 3. Participatory Augmented Reality Business Card Information System

The demand for business communication linked to the real world using analog sensibility has grown. Services utilizing QR codes on paper business cards have emerged to cater to this need. However, such methods provide content designated by the user by redirecting to a site linked to the QR code, resulting in limited interaction between the content and the user and, ultimately, a less engaging experience.

Business cards that employ Augmented Reality (AR) technology offer a more interactive experience by providing 3D animations, 2D videos, sounds, and other integrated content within the real-world context. These AR business cards facilitate brand and user promotion and ease any initial awkwardness in offline meetings. They enhance traditional paper business cards by adding visual information and engaging features and automatically generating buttons that link to phone numbers, location information, and websites. This approach enables various interactions for users.

AR business cards can be converted into digital business cards through a camera, allowing users to form business-centric community platforms, e.g., by exchanging business cards and engaging in chats by registering and managing business card information. This approach offers high scalability.

Furthermore, the business market using AR is expanding, and AR-based business card systems have become increasingly marketable. It is anticipated that such techniques will integrate with various offline printed materials. Since many businesses already employ post-processing and multiple options for efficient transactions with business cards, creating AR-based business card systems could enable efficient product promotion without incurring additional marketing costs.

To invigorate Augmented Reality (AR) products, providing exceptional content that caters to customer needs is crucial. Developing emotionally engaging, high-quality content templates based on storytelling is vital for promoting the widespread adoption of AR content. As diversity and a sustainable, regular content supply should be prioritized, this research aims to explore efficient AR business cards that can enter the market.

This study seeks a content creation template that enables users to develop various AR content tailored to their business cards. The objective is to design a user-participation business card content provisioning service and operating system which enhances and delivers content matched with business cards through mobile devices.

Figure 4 illustrates the AR business card system’s business model, the interaction between customers and business card providers, and the content of offering paper business cards and AR. Figure 5 depicts the implementation of the participatory AR business card information system, showcasing the necessary components for each module interaction.

The advanced business card information provisioning service design encompasses the following five technical areas:Cross-platform-based system environment and efficient interface:

The design should enable the transmission and reception of pattern information, content resources, and template information through a web server. It should also allow business card management and content provision regardless of location. The interface layout should be designed as a Multi-Document Interface (MDI) to accommodate complex tools.


2.Development of pattern detection and storage technology:


A unique pattern must be applied to offline business cards and stored in a database, allowing for the detection of the business card’s unique ID based on stored pattern information.


3.Development of user-participation content enhancement technology:


Users should be able to create extended objects by combining multiple content types based on the content production template. The extended objects should support various interactions, including 3D models, 2D images/videos, animation buttons, and URL links.


4.Development of comprehensive content resource registration technology.


In this research, a viewer can provide 2D content through streaming services. A template can be created by arranging 3D content in a virtual space, specifying the content created by the 3D modeling operation as an Asset, and storing the virtual coordinate information of the object file so that the user can retrieve it from the content template when creating an enhancement object. Then, we developed technology that makes it possible to link with mash data.


5.Development of integrated content generation technology:


Instead of augmenting simple characters or 2D images, this research aims to develop a template that allows users to synthesize desired content and generate a single enhanced object. The technology should enable users to create extended objects by inserting various content into the template.


6.Development of integrated content resource registration technology:


This technology should allow viewers to provide 2D content through streaming services and create templates by arranging 3D content in a virtual space. Users should be able to call upon the content created by the 3D modeling operation as an Asset when creating an enhanced object. The virtual coordinate information of the object file should be passed in XML tag format to enable linkage with mash data.

## 4. Location Information Provision Service Module

As illustrated in Figure 6, the location information provision service employed in this research utilizes the address information from the business card holder’s data stored in the business card information database. This is carried out sequentially by the location information provision service module. The process involves obtaining location information and using the GPS stored on the smartphone to determine the user’s location.

The module employs a series of steps, including route calculation, map information verification, user position display, business card holder position display, and real-time navigation assistance. The route calculation step computes a path from the user’s position to the address of the business card holder using advanced routing algorithms. This ensures that the most efficient route is selected, considering real-time traffic data and other factors that might impact travel time.

The map information verification step involves transmitting map information containing the calculated route from the server and confirming it. This allows the user to preview the suggested route and adjust it if needed, e.g., by adding intermediate stops or avoiding specific areas.

The user position display step shows the user’s location on the transmitted map, providing real-time updates as they move. This ensures that users can easily track their progress along the route, make necessary adjustments, and access their current position at all times.

In contrast, the business card holder position display step presents the business card holder’s location on the transmitted map using the 3D face model of the business card holder stored in the 3D object database. This unique feature enhances the user experience by providing a personalized representation of the destination, making it easier to recognize and remember.

The real-time navigation assistance step guides the user along the calculated route using audio and visual cues, ensuring a seamless and efficient journey to the business card holder’s location. The system can also dynamically adjust the route based on changing traffic conditions or user preferences, offering a truly adaptive navigation experience.

This comprehensive approach ensures accurate and user-friendly location information for enhanced business communication and networking. By integrating advanced mapping technology, personalized visualizations, and real-time navigation assistance, the location information provision service offers a powerful tool for professionals looking to strengthen connections and facilitate in-person meetings in an increasingly digital world.

## 5. Discussion

This research has demonstrated the feasibility of implementing cost-effective pricing policies by adopting a low-cost approach to content production, system maintenance, and upkeep. The developed system enables users to generate augmented objects tailored to their specific business content needs, leveraging the existing business card market infrastructure, including QR codes and mobile business cards.

By prioritizing quality control and offering superior content compared to traditional Augmented Reality business card systems, this research presents a distinct technical advantage that can significantly enhance the user experience. This innovation is anticipated to promote the widespread adoption of Augmented Reality business card services, with consumers taking on more engaged roles as content creators rather than merely passive consumers.

The system developed through this research exemplifies a pioneering approach to business card systems utilizing Augmented Reality technology. It provides diverse business content through paper business cards, empowering users to create customized content tailored to their business meetings or marketing strategies. This innovative approach effectively addresses the contemporary business card market’s needs, offering considerable potential for widespread adoption and commercial success.

Furthermore, by emphasizing quality control at every stage of the AR business card development process, the system ensures a consistently positive user experience. This focus on quality control and the system’s cost-effective and user-friendly nature can revolutionize the business card industry and set new standards for future developments in Augmented Reality applications.

This research has explored an inventive, user-centric approach to AR business cards that prioritizes quality control and affordability. By providing a seamless, engaging experience that meets the demands of the contemporary market, the system holds great promise for transforming how business professionals connect and communicate in the digital age.

## 6. Conclusions

Table 1 compares traditional and augmented reality business cards, emphasizing the role of quality control in developing and adopting such innovations. Information and communication technology advancements have recently allowed for the incorporation of Augmented Reality (AR) technology into smartphones, leading to the development of AR-enhanced business cards featuring 3D graphics. This research focuses on applying AR technology to business cards, often serving as the first point of contact in marketing and business endeavors. It examines the role of quality control in maximizing their effectiveness.

A crucial aspect of realizing AR in business cards is marker recognition technology for image marker detection, including pre-processing technology for marker detection and pattern matching technology for marker pattern detection. Ensuring quality control during these processes is vital for providing a reliable and engaging AR experience. Moreover, user content control necessitates advanced object control technology that guarantees a seamless AR experience while maintaining high quality.

Additionally, this research supports location information services that enable users to obtain a business card holder’s location using address information stored in the business card information database. Quality control ensures accurate and reliable location information, enhancing the user experience.

To promote the widespread adoption of AR content, this research has developed an emotionally engaging, high-quality content template based on storytelling. By maintaining strict quality control standards, users can generate a range of AR content tailored to their business cards through a content creation template. Recipients of these AR-enhanced business cards can access and interact with the augmented content via mobile devices, ensuring a memorable and engaging experience.

This research’s business implications and anticipated effects include increased user loyalty through appropriate discounts on selected AR business card content and the option for content upgrades, all while maintaining quality control. This approach can enhance content provision, maintenance, and sales management. Furthermore, developing a business concept character within the AR business card content template theme allows for the creation of a 3D animation series, enabling expansion into collaborative businesses such as production, webtoons, and virtual reality content.

In conclusion, this research has designed and explored an innovative, user-centric approach to AR business cards with a strong focus on quality control. This emphasis on quality control paves the way for enhanced marketing and business communications in the digital age, ensuring an engaging, effective, and memorable experience for users.

## Figures and Tables

**Figure 1 sensors-23-04068-f001:**
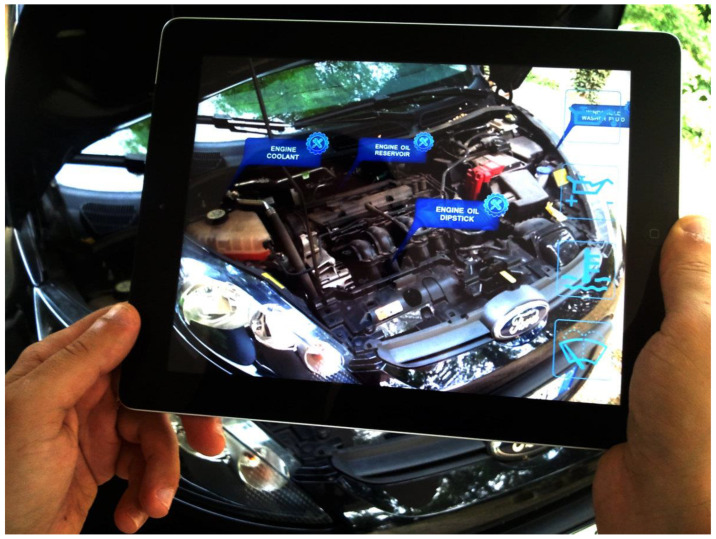
Scene From Ford’s AR repair help software [15].

**Figure 2 sensors-23-04068-f002:**
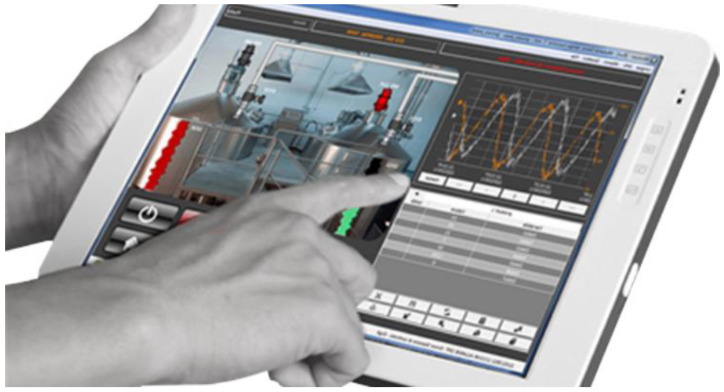
Movicon’s CE Overview [16].

**Figure 3 sensors-23-04068-f003:**
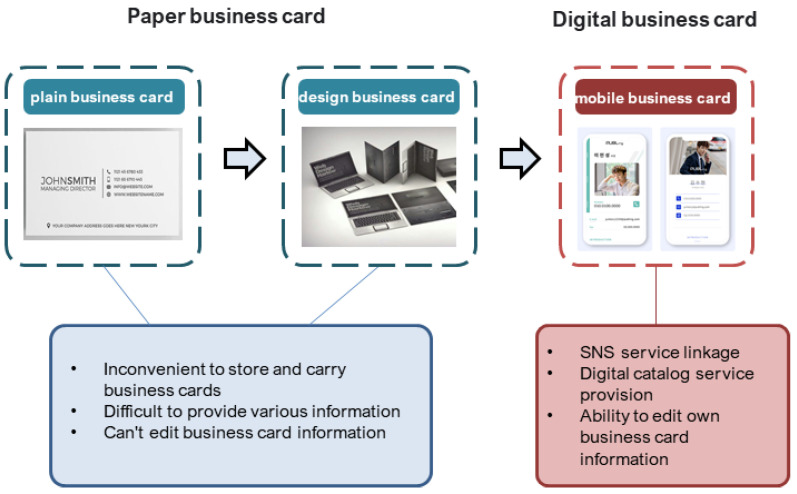
Paper business cards and digital business cards.

**Figure 4 sensors-23-04068-f004:**
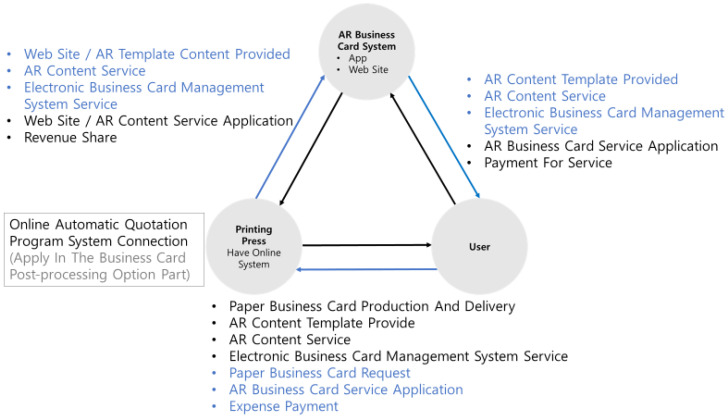
Augmented Reality business card system business model.

**Figure 5 sensors-23-04068-f005:**
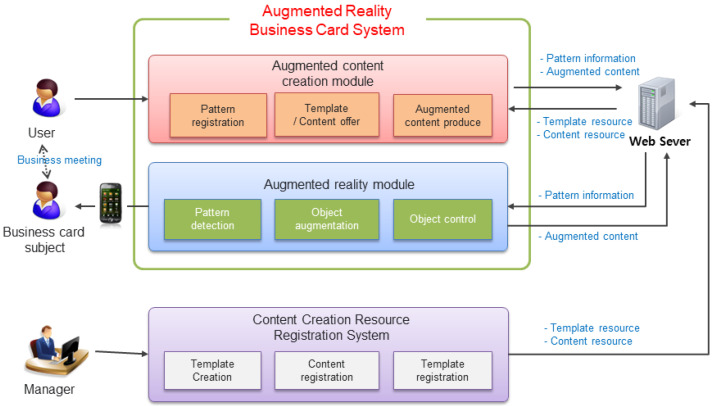
Configuration diagram of participatory Augmented Reality business card information system.

**Figure 6 sensors-23-04068-f006:**
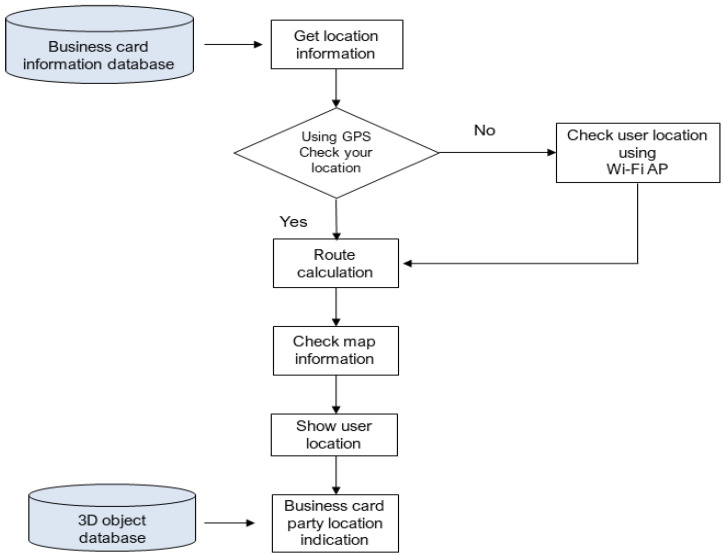
Location information provision service flow chart.

**Table 1 sensors-23-04068-t001:** Comparison of Traditional Business Cards and Augmented Reality Business Cards.

Feature	Traditional Business Card	AR Business Card
Information presentation	Static text and images	Dynamic 3D graphics and animations
Content interaction	None	Interactive with smartphone
Location services	None	Enabled Through the business card information database
Customization	Limited to text and images	Multiple AR content templates, character creation, animations
Engagement	Limited	High amount of storytelling and emotional connection
Business Expansion Opportunities	Limited	Collaboration potential in animation, webtoons, and virtual reality content

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
