# Peer review of "Enhancing Quality Control in Web-based Participatory Augmented Reality Business Card Information System Design"

_sensors, 2023, doi:10.3390/s23084068_

Round 1

Reviewer 1 Report

The idea of AR business cards is very good. The system description is pretty clear and well-described. However, it is unclear whether the system is implemented or not. The authors mention "a content creation template", but they do not make it available to be seen and thus understood if the idea is feasible.

I would recommend expanding the paper with the implemented template and conducting a user experience evaluation study to verify user satisfaction.

Some minor remarks:

The AR technology description is too detailed and should be shortened. If possible, find and describe similar publications about using AR instead of printed information.

- proofread the paper by an English-language professional

Pg. 5 the Section 2.2. should not start with "In addition"

line 190: first person expressions are not used in scientific publications, change to "we could not..."

Reviewer 2 Report

This paper focused on the design of an AR business card system. I was surprised to not see any references related to business cards however. There is research on business cards that would support the early claims and possibly even some that address the current use of AR. 

Line 97-98 begins with verb, the noun is missing from the sentence.

3. Participatory Augmented Reality Business Card Information System

This section seems less complete than the previous sections and provided a lot more unsupported details or ideas rather than a cohesive problem and explanation of the project and study.

There were several incomplete, odd, or poorly worded sentences in this section, here are a few:

Line 190 – “I couldn’t experience it.” Seems like a mistyped or placed sentence?

Line 195-196 – this is a run on sentence, split into two.

Line 201 – “Highly scalable” isn’t a complete sentence.

Line 248 – “It has a combination of functions that can.”  Can what? This is an incomplete sentence.

Lines 204-208 – this seems to be several incomplete or run-on sentences. It is unclear what is being said here.

It seems from Figure 5 (which needs work as well, the text between the bubbles needs to be larger as it is harder to read at 100%) indicates the model that the proposed study is addressing, but there was no citation in the prior paragraphs to indicate the problem clearly.

What is meant on Line 212-213 by “creating emotional high-quality content templates based on storytelling is essential...”? What is emotional here and how do you incorporate story-telling? It seems you are asking that to come directly from the users through the templates – but there is no explanation for how the templates do this. A bit more description of the template is necessary to support this claim as much of your technical explanations are on how to get the technology to accept and interact with the different media and AR registration system.

Starting at line 217, this is where section 3 becomes a little clearer.

A suggestion given the first paragraphs of section 3 are meant to point out the nuanced needs of the business card system is to provide a table (and include citations) for the specific needs so when you provide the 5 technical areas (starting line 230) it is clear why specific decisions were made and how they compare to the existing AR Business Card system – making this a little easier for a reader to see the advantages you later mention exist by section 5.

Figure 6 is easier to read at 100% than Figure 5 but increasing the contrast of the text a little more will help readability.

What about this project was a study and/or participatory? There was a note about a technical area to create a user-participation content enhancement technology but why? Is this what makes the system participatory? – it seems like the idea was designed (and maybe tested) based on Section 5 though it is unclear.

Section 4.

Lines- 270-276 – is all one sentence. Given the use of punctuation changes from a list to separate clauses, please use multiple sentences instead.

Section 5.

Line 290 – “Can effectively enter the business card market.”  - what is the subject of this sentence?

Section 6 Conclusion

Overall, I think the paper touches on an interesting topic, there are some issues with seeing the whole project as it is presented though. I don’t really see templates in what was written though I do see a process of media integration being more streamlined to enable AR registration to occur more easily. If this process is the “template” I then do not see how it is considered “emotional high-quality content” because something like that would be entirely dependent on the user – not the system. The storytelling aspect of the template is not as well explained though my interest is piqued as to how it would be achieved in such a system to assist the broader public in creating these types of AR business cards.

There were a lot of writing issues in the later half of the paper which made it harder to interpret the information as well. I pointed out several grammatical issues though not all of them – a good proof-reader would help to better communicate all the work that was clearly done in this project.

For the “study” portion of the project, it was hinted that the new systems improves on the existing AR business card system – I would really like to see the two next to each other in comparison as I noted in my earlier comments. A table or diagram to illustrate the improvements would be incredibly helpful in supporting the claims made in section 6 Conclusion.

Round 2

Reviewer 2 Report

The revisions add much to the previous version of this manuscript. The suggestion to add references on paper business cards however was not adequately met. Wikipedia is not a reliable source of information for citing about the use and nature of business cards. 

While I do not do research in this area specifically with business cards, there is research out there about their design and purpose. Even the Wikipedia page sited will have references that can be used in lieu of Wikipedia directly.

Google Scholar is another good place to search as well.

The figures and tables are very much improved and make more sense of what you actually did. The English language revisions were also good.
